# Comprehensive characterization of *RAS* mutations in colon and rectal cancers in old and young patients

Ilya G. Serebriiskii[1,2], Caitlin Connelly[3], Garrett Frampton[3], Justin Newberg[3], Matthew Cooke[3], Vince Miller[3], Siraj Ali[3], Jeffrey S. Ross[3,4], Elizabeth Handorf[5], Sanjeevani Arora [6], Christopher Lieu [7], Erica A. Golemis [1] & Joshua E. Meyer [1,8]

Colorectal cancer (CRC) is increasingly appreciated as a heterogeneous disease, with factors such as microsatellite instability (MSI), cancer subsite within the colon versus rectum, and age of diagnosis associated with specific disease course and therapeutic response. Activating oncogenic mutations in *KRAS* and *NRAS* are common in CRC, driving tumor progression and influencing efficacy of both cytotoxic and targeted therapies. The *RAS* mutational spectrum differs substantially between tumors arising from distinct tissues. Structure-function analysis of relatively common somatic *RAS* mutations in G12, Q61, and other codons is characterized by differing potency and modes of action. Here we show the mutational profile of *KRAS*, *NRAS*, and the less common *HRAS* in 13,336 CRC tumors, comparing the frequency of specific mutations based on age of diagnosis, MSI status, and colon versus rectum subsite. We identify mutation hotspots, and unexpected differences in mutation spectrum, based on these clinical parameters.

[1] Program in Molecular Therapeutics, Fox Chase Cancer Center, Philadelphia, PA 19111, USA. [2] Kazan Federal University, 420000 Kazan, Russia. [3] Foundation Medicine Inc., 150 Second Street, Cambridge, MA 02141, USA. [4] Upstate Medical University, Syracuse, NY 13210, USA. [5] Bioinformatics and Biostatistics Facility, Fox Chase Cancer Center, Philadelphia, PA 19111, USA. [6] Program in Cancer Prevention and Control, Fox Chase Cancer Center, Philadelphia, PA 19111, USA. [7] Division of Medical Oncology, University of Colorado Cancer Center, Aurora, CO 80045, USA. [8] Department of Radiation Oncology, Fox Chase Cancer Center, Philadelphia, PA 19111, USA. Correspondence and requests for materials should be addressed to J.E.M. (email: joshua.meyer@fccc.edu)

C olorectal cancer (CRC) is the fourth most common cancer in the United States, with an estimated 135,430 new cases diagnosed in 2017 and 50,260 deaths in the same year[1]. While the overall survival (OS) at 5 years is 65%, in patients with metastatic disease this rate drops to 14%[1]. Ongoing efforts to improve diagnosis and treatment for CRC depend on more accurate understanding of disease biology. For CRC, as for many cancers, a disease once considered as uniform is being increasingly divided into discrete subclasses, based on the integration of molecular and clinical analyses. For example, microsatellite stable (MSS) CRC tumors have distinct etiology and treatment recommendations from CRC with a high level of microsatellite instability (MSI-H)[2]. Other potentially important distinctions are emerging, such as formation of tumors on the right or left side of the colon[3,4].

While typically grouped together, cancers arising in the colon differ from those arising in the rectum in several clinically important ways. First, early-stage patients with colon cancer demonstrate improved survival over rectal cancer in stage IIB, while in stages IIIC and IV the opposite is true[5,6]. Interestingly, although the incidence of CRC has been increasing over time, particularly among young patients, the increase has been more notable in rectal cancer[7,8]. Second, rectal cancer is treated differently from colon cancer, with preoperative radiation with or without chemotherapy required to achieve acceptable locoregional control in rectal cancer[9]. Because rectal cancer is less common than colon cancer (28% of total CRC cases in the United States), data sets to analyze molecular differences have been limited[10]. Further complicating the analysis, the natural history of CRC appears different in old versus young patients, as evidenced by increased lymph node metastasis rates in younger patients, possibly suggesting a difference in biology[7]. To date, examinations of age of onset as a factor influencing molecular differences in CRC have also been limited, with the notable exception of MSI. MSI is more commonly found in younger patients, potentially due to the activity of inherited damaging variants in genes associated with DNA repair, which tends to lead to an early-onset disease[11]. Additionally, one recent publication has shown increased alteration of TP53 and CTNNB1 in youngers patients, with older patients demonstrating increases in APC, KRAS, BRAF and FAM123B alterations[12].

The three RAS GTPases (KRAS, NRAS, and HRAS), and particularly the family member KRAS, are the most commonly mutated oncogenes in cancer[13]. KRAS and NRAS have been reported to have activating missense mutations in ~40 and 3–5% of CRCs, respectively[14,15]. Both preclinical and clinical data show that the specific codon mutated, and the sequence to which it is mutated, differentially impact the activity of the protein, the prognosis, and the response to epidermal growth factor receptor (EGFR)-directed therapy in patients with CRC[16–22]. In this study, we describe the frequency of missense mutations and copy number variation (CNV) of wild-type (wt) and mutated alleles of KRAS and NRAS in a very large dataset of 13,336 CRC tumors from patients profiled by Foundation Medicine (FM) Inc. We separately report results based on occurrence in MSS versus MSI-H tumors, in colon versus rectal cancer, and based on patient age. These data indicate non-equivalent profiles of specific CRC-associated mutations based on subsite, age, gender, and other criteria.

## Results

**Patient population: age, gender, tumor site, and MSI status.** Comprehensive genomic profiling (CGP) of tumor samples by next-generation sequencing was performed in the course of routine clinical care for CRC patients with advanced disease,

**Table 1 Clinical characteristics of 13,336 colorectal cancer patients in study**

| Site | Number | % |
|---|---|---|
| Colon | 11,600 | 85.30 |
| Rectum | 2003 | 14.70 |
| Sex | | |
| F | 6137 | 45.10 |
| F-discordant | 2 | 0.01 |
| M | 7453 | 54.80 |
| M-discordant | 11 | 0.08 |
| Microsatellite | | |
| MSI-H | 455 | 3.3 |
| MSS | 9615 | 70.7 |
| Unknown/ambiguous | 3533 | 26.0 |
| Age | | |
| Mean | 56.96 | |
| SD | 12.49 | |
| Median | 57 | |

MSS microsatellite stable, MSI-H microsatellite instability high. Individuals with rectal cancer were slightly younger than colon cancer patients (Supplementary Fig. 1). Tumors with discordant sex were not included in the analysis

resulting in the identification of genetic variants as well as biomarkers such as tumor mutation burden (TMB) and microsatellite stability. Characteristics of the patient population are summarized in Fig. 1a, Table 1, and Supplementary Fig. 1. The dataset analyzed here is nearly 2-fold greater than the combined previously analyzed patient cohorts reported in publications[23,24] or developed through the GENIE consortium[25], or otherwise available at cBioPortal. Comparable to other publicly available sequencing data (PAD) for CRC, the group is 45.2%:54.8% female:male, with a similar age distribution at the time of sequencing, and a similar proportion of 85.3% colon:14.7% rectum.

Among the total set of 13,336 CRC tumors, information on microsatellite stability and quantifying TMB was available for 10,070 (Fig. 1b), of which 9615 (95.5%) were MSS and 455 displayed MSI-H (4.5%). While information for neither MSS/MSI-H nor TMB status was available for 267 patients, information on TMB status only was available for 3266 patients. Using the set of 10,070 tumors as a training set, we determined a threshold of <16 mutations separated 99% of the MSS from 99% of the MSI-H cases (Supplementary Fig. 2). Applying this threshold, we assigned 3096/3266 (95%) additional tumors as likely MSS, and 170/3266 (5%) as likely MSI-H. Combining these specimens with those with known MSS or MSI-H status, we designated 12,711 tumors as MSS/low TMB (subsequently designated as MT-L) and 625 cases as MSI-H/high TMB (subsequently designated as MT-H). Age 40 years has been used to examine the increasing incidence of CRC in young people, due to the lack of CRC screening below age 40 years, thereby removing screening as a source of bias[26–28]. For some analyses, we chose this as our dichotomous cut-point for age.

In initial control analyses, we benchmarked frequency of MT-H, MSI-H, and TMB-H CRC (Fig. 1c–e and Supplementary Fig. 3) based on age, gender, and tumor subsite. Results were essentially equivalent among the three groups. MSI and TMB were higher in younger patients, based almost entirely on elevated levels in young males (Fig. 1c, Supplementary Fig. 2). MSI and TMB were also significantly elevated in colon versus rectal tumors, regardless of gender (Fig. 1d, Supplementary Fig. 2), but correlated with younger age (Fig. 1e, Supplementary Fig. 2). Given these similarities, subsequent analyses were based on the larger MT-H versus MT-L cohorts.

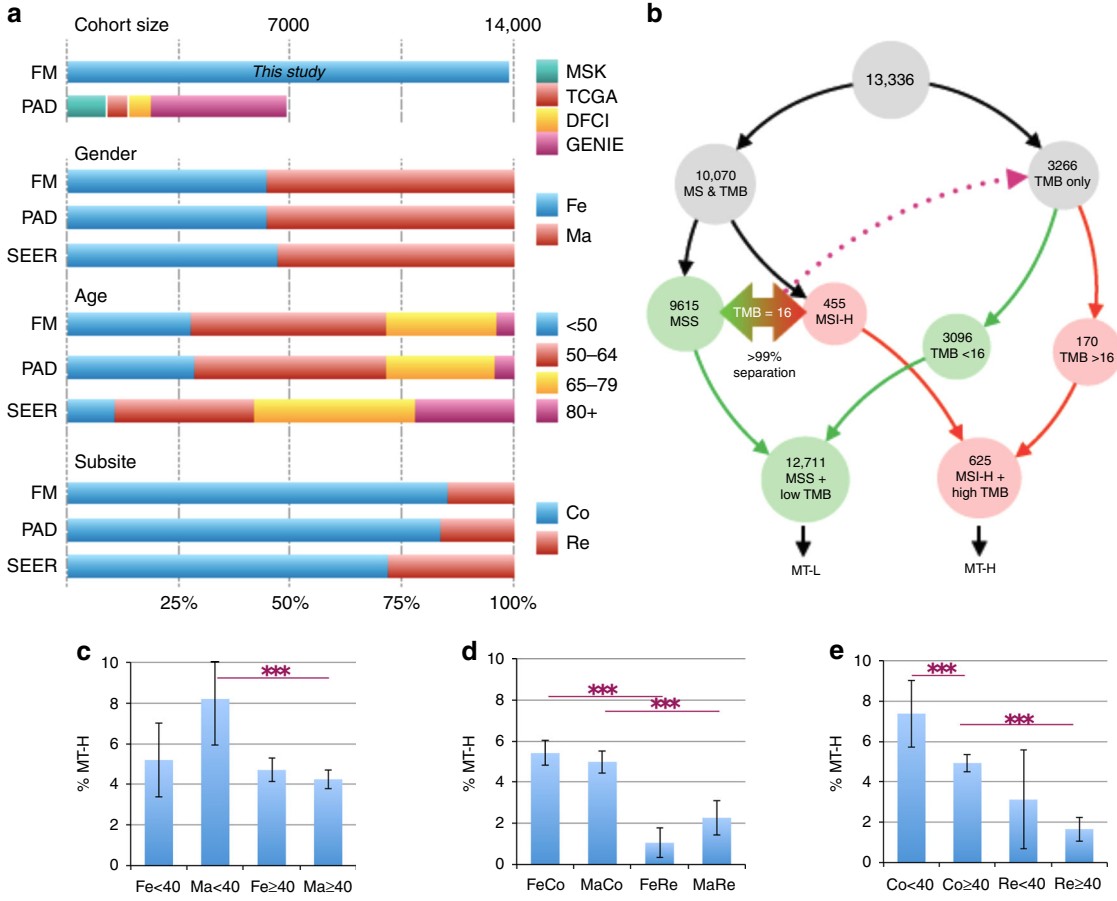

**Fig. 1** Overall profile of dataset. **a** Comparison of Foundation Medicine (FM) Inc. dataset in the present study versus a benchmark group of publicly available data (PAD) for colorectal cancer (CRC) published by Memorial Sloan-Kettering (MSK)[24], the Dana Farber Cancer Institute (DFCI)[23], the Genomics Evidence Neoplasia Information Exchange (GENIE)[25], and The Cancer Genome Atlas (TCGA); population characteristics are also compared to overall population reported in SEER (Surveillance, Epidemiology, and End Results)[1]. **b** Flowchart and analysis tree for populations defined by FM as high level of microsatellite instability (MSI-H) or microsatellite stable (MSS), or with known tumor mutation burden (TMB); generation of MSI-H/high TMB (MT-H) and MSI-H/low TMB (MT-L) analysis cohorts. **c–e** Frequency of MT-H tumors based on parameters of gender (Fe, female or Ma, male) as a factor of age (<40 or ≥40 years) (**c**), or primary tumor site (Co, colon or Re, rectum) (**d**); or in combined genders, based on age and tumor site (**e**). Error bars, 95% confidence intervals. Statistical significance is denoted by ***$p < 0.005$

**KRAS, NRAS, and HRAS mutation hotspots in the entire cohort.** Among the 13,336 tumors analyzed (Supplementary Data 1), there were 6926 *KRAS* alterations (defined as cumulative missense mutations, frameshifts, and copy number changes), including 6634 missense mutations, in the MT-L cohort of 12,711 patients, and 246 *KRAS* alterations (235 missense) among 625 MT-H TMB patients (Supplementary Data 1). Thirty-four alterations, including 32 missense mutations in *NRAS*, were detected in MT-H patients, and 560 alterations (537 missense mutations) in MT-L patients (Supplementary Data 1). For *HRAS*, 35 missense mutations (44 alterations in total) were identified among MT-H patients, and 32 missense mutations (53 alterations in total) among MT-L patients.

The complete set of missense mutations in *KRAS*, *NRAS*, and *HRAS* were then analyzed to identify mutational hotspots (Fig. 2a–c), and mapped to the protein structure for each *RAS* protein (Fig. 2d–k). This identified 16 mutational hotspots for *KRAS* (Fig. 2a); these included four sites previously reported from a dataset of >1267 CRC tumors[29], which were a subset of 10 hotspots identified from an analysis of ~25,000 tumors of all types[30], as well as 6 additional sites not previously apparent in analysis of smaller cohorts of CRC tumors (residues V9, V14, R68, R164, K176, and K180). For *NRAS* (Fig. 2b), this analysis identified one new hotspot for *NRAS* mutation (residue P185), in

addition to three previously defined hotspots (residues G12, G13, and Q61). For *HRAS*, no new hotspots were unequivocally identified, although two residues (R164 and P167) were borderline significant (Fig. 2c). None of the identified mutations were associated with microsatellite regions.

RAS functions as a molecular switch between the GTP-bound active form and the GDP-bound inactive form. Missense mutations in RAS proteins alter the homeostatic balance between these two forms, either by reducing GTP hydrolysis or by increasing the rate of GTP loading[31]. The most common *KRAS*-activating mutations, in codons 12, 13, 61, 117, and 146, cluster around the nucleotide-binding pocket[16]. Display of the full set of mutational hotspots from *KRAS* on the three-dimensional (3D) structure of the KRAS protein (Fig. 2d–g) shows the majority of these clusters in functional domains. One new hotspot (V14) is located within 5 Å of K117; others (e.g., K176, K180) are localized within the C-terminal hypervariable region (HVR). Finally, our analysis did not identify two known hotspots in *KRAS* (residues K5 and A11), which were described in non-CRC tumors, reflecting known tissue bias in hotspots[30]. Interestingly, enrichment for mutations was observed within two short spans of six amino acids, although not targeting individual residues within the spans. This includes residues 47–52, which encompasses 10 mutations ($p = 7.2^{E-03}$), and positions 62–67 (between the

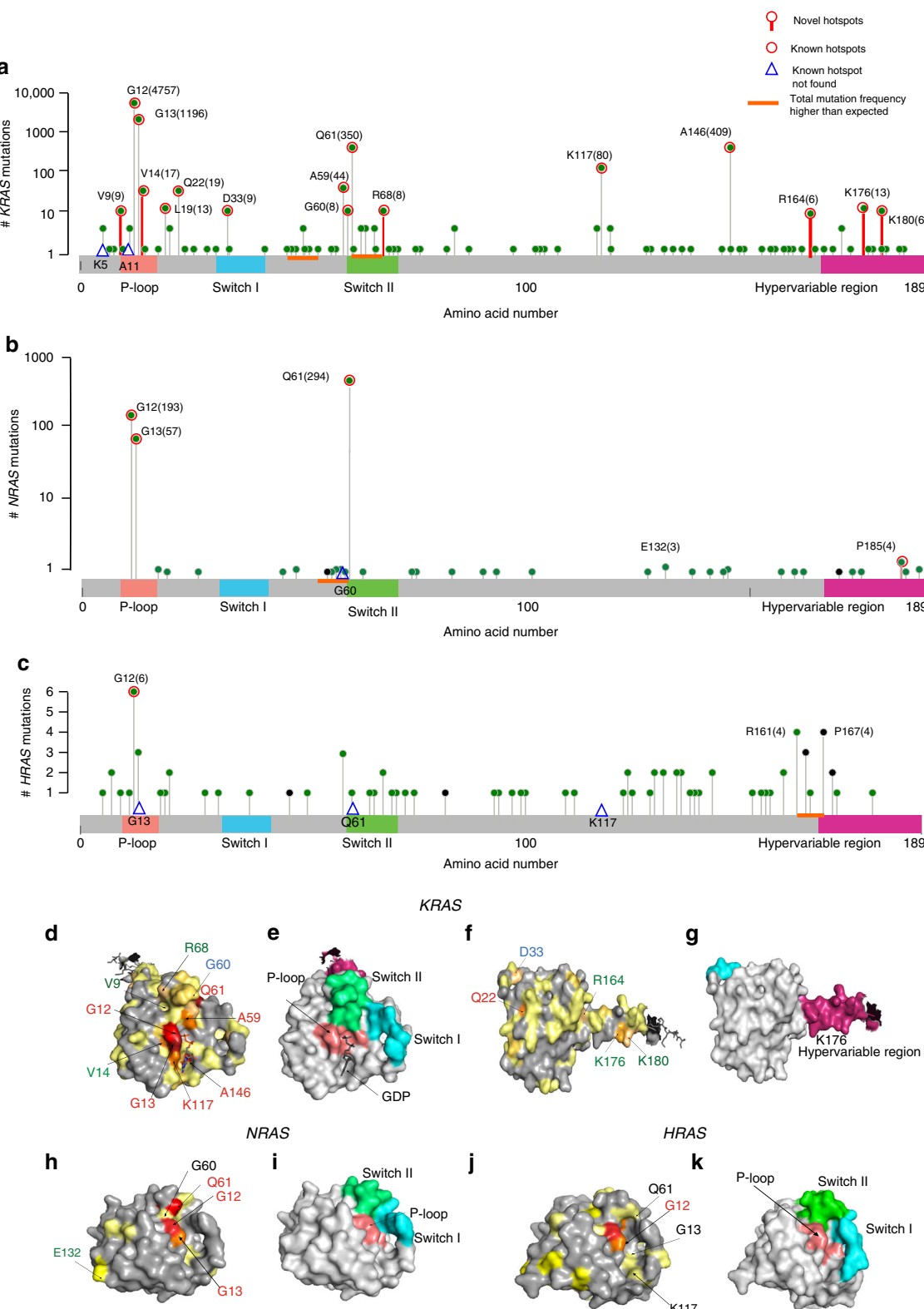

known Q61 and new R68 hotspots), which encompasses 11 mutations ($p = 2.7^{E-04}$).

Display of the full set of mutational hotspots from the 3D structure of *NRAS* (Fig. 2h, i) shows that the majority of these cluster in functional domains. The new hotspot (P185) is localized within the C-terminal HVR. In addition, a marginally

significant potential hotspot (E132; three mutations, $p = 0.018$) is located in the α-helix between two amino acid motifs (G4, NKXD and G5, EXSAK) that are important for binding the guanine base of the nucleotide[32]. *KRAS* does not have hotspots comparable to E132 and P185 of *NRAS*. The G60 hotspot, identified in other cancer types[30], was not found in our study;

**Fig. 2** Visualization of *RAS* mutation burden, hotspots, and functional domains. **a–c** Top graphs indicate mutation burden per codon of the *RAS* proteins *KRAS* (**a**), *NRAS* (**b**), and *HRAS* (**c**), respectively; for *KRAS* and *NRAS*, the data are plotted on a logarithmic scale. Red circle indicates missense hotspots (new and previously reported); red stem, hotspot identified in this study. Total number of mutations is provided within parenthesis. Blue triangle, previously reported hotspot not found in this study. Orange bars indicate regions where total mutation frequency is higher than expected. Critical functional domains are indicated; P-loop (salmon), switch I (cyan), switch II (green), and hypervariable region (pink). **d–k** Alignment of mutations to protein structure for *KRAS* (**d–g**), *NRAS* (**h–i**), and *HRAS* (**j, k**), visualized by PyMOL (http://www.pymol.org). **d, f, h, j** Font color indicates new hotspots (green), known colorectal cancer (CRC) hotspots (red) detected in this study, and other known hotspots not found in this study (black). **d–g** GDP-bound full-length *KRAS* structure[59] (PDB code: 5TAR); two views (**d, e** versus **f, g**) of the folded protein, indicates mutations (**d, f**) and functional motifs (**e, g**). **h, i** *NRAS* structure[32] (PDB code: 5UHV) indicates mutations (**h**) and functional motifs (**i**). **j–k** *HRAS* structure[32,60] (PDB code: 1CTQ), indicates mutations (**j**) and functional motifs (**k**). Color key: **d, f**, never mutated, gray; 1–5 times, pale yellow; 6–10 times, yellow-orange (residues 180, 164, 68, 60, 33, and 9); 11–15 times, light orange (residues 19 and 176), 16–20 times, tv_orange (residues 14 and 22); 20–100 times, orange (residues 59 and 117); > 100 times, tv_Red (residues 61, 146, and 13); and >4000 times, firebrick (residue 12). **h** Never mutated, gray; 1–2 times, pale yellow; 3–5 times, yellow (residues 132 and 185; 185 not shown, 5UHV lacks hypervariable region); 59 times, orange, (residue 13); 204 times, tv_Red (residue 12); 306 times, red (residue 61). **j**, never mutated, gray; 1 time, pale yellow; 2 times, yellow (residues 7, 20, 68, 123, 129, 131, 134, 135, 158, and 169; 169 not visualizable, 1CTQ lacks hypervariable region); 3–4 times, orange (residue 13, 59, 161, 163, and 167; back view is not visible); 6 times, tv_Red (residue 12)

however, mutations were enriched overall within the five amino acids immediately upstream of Q61 (amino acids 56–60, seven mutations, $p = 5.1^{E-04}$) (Fig. 2b).

For HRAS, of the four hotspots previously nominated based on a pan-cancer survey (G12, G13, Q61, K117;[30]), only one (G12) was also identified in our study (Fig. 2j, k). However, two previously overlooked residues (R161 and P167) were mutated at a higher frequency than expected (four mutations each; $p = 0.007$), and an additional four mutations were identified in residues 163 and 164. These above mutations are unusually enriched ($p = 2.0^{E-04}$) in a stretch of amino acids located near the N terminus of the HVR (Fig. 2c). The HVR is distinct between the RAS isoforms and this unusual enrichment of mutations in the N terminus of this region in the HRAS protein may play a role in disease pathogenesis.

**Mutational variation between MT-L and MT-H tumors.** We compared the distribution of total alterations in *KRAS*, *NRAS*, and *HRAS* in CRC patients, as a factor of MT-L versus MT-H status (Fig. 3a). *KRAS* mutations were higher in MT-L than MT-H CRC. For *NRAS*, no significant differences in the total number of mutations were noted between MT-L and MT-H CRC tumors. Surprisingly, even though the total frequency of *HRAS* mutations was very low in CRC, the frequency of *HRAS* mutations in MT-H tumors was significantly higher than in MT-L tumors, comparable to that of *NRAS* mutations. This pattern differed significantly from the pattern observed for *KRAS* (where the mutation frequency was higher in MT-L CRC) or *NRAS* ($p < 2.2^{E-16}$).

The observed spectrum of *KRAS* missense mutations also significantly differed between MT-L and MT-H tumors ($p = 5.0^{E-04}$) (Fig. 3b). G12 mutations predominated in MT-L tumors (69.3%), whereas MT-H tumors had a greater distribution of sites targeted, including G12 (~34%), G13 (~35%), A146 (~9%), and other missense structural variants (SVs) (~22%). Among the G12 mutations, statistically significant differences in codon substitution preference were also observed, with over 77% of MT-H tumors having G12D mutations and no other specific codon substitution comprising an allele fraction >8%, versus MT-L tumors having ~44% G12D, ~31% G12V, ~11% G12C, ~6% G12A, and ~8% other G12 variants ($p = 2.4^{E-08}$) (Fig. 3c). This difference likely reflects the fact that G12D changes result from C > T nucleotide changes, which are known to be enriched in MT-H tumors[33]. Variance in patterns of codon substitutions was also observed with other codons (e.g., A146), although with lesser significance due to smaller sample size.

In contrast to the lack of difference in total frequency of mutations, the *NRAS* mutational spectra differed significantly

between MT-H and MT-L tumors ($p = 3.2^{E-13}$) (Fig. 3b). MT-H tumors were dominated by a large number of low-frequency mutations (~53%), followed by G12 (25%), Q61 (~16%), and G13 (~6%). The most common site of missense mutation in MT-L tumors was Q61 (~52%), with G12 (~33%), G13 (~10%), and other variants (~4%) making up the remainder of single codon substitutions. Due to the small sample size for *NRAS* alterations, further analysis of specific missense variants was not conclusive, with the exception of the most common variant, G12. In MT-L tumors, over 80% of 178 mutations affecting G12 are the substitutions G12A, G12V, and G12D, while in 8 MT-H tumors, 7 are substitutions to other residues, with G12C being predominant (5/8 samples, compared with only ~12% in MT-L (Fig. 3c).

The number of *HRAS* missense mutations observed in this dataset was too low for a significant analysis of mutation preference.

**Missense mutational frequency and spectrum dependent on age.** We analyzed *RAS* mutations in CRC patients as a continuous variable of age. In MT-L patients, the overall frequency of *KRAS* alterations increase with age ($p = 3.1^{E-03}$, Fig. 3d). In sharp contrast, the incidence of *KRAS* alterations is reduced with age in MT-H patients ($p = 2.1^{E-03}$, Fig. 3d). Among the MT-L patients, the frequency of missense variants in *NRAS* remained almost constant with age (Fig. 3d). For MT-H *NRAS* mutations, and for all *HRAS* mutations, there were insufficient numbers to identify significant age-related trends.

We next compared the profile of specific *KRAS* mutations based on the age of the patient at the time of sequencing. This was done both by considering age as continuous variable and by using age 40 as a stratification point in individuals from 8 to >85 years of age. Analysis of the MT-L cohort by both approaches revealed that the most significant difference in mutational spectra of younger and older patients was the greater incidence of G12 mutations prevalent in younger patients versus the more frequent incidence of A146, K117, and Q61 mutations in older patients, while the incidence of G13 remained the same ($p = 1.3^{E-03}$) (Fig. 4a, b). Among the G12 mutations (Supplementary Fig. 4a, b), the frequency of substitution to G12V fraction appears to slightly increase with age, while substitution to G12A and G12C decreases. Strikingly, the overall fraction of Q61 substitutions in the *KRAS* mutation pool more than doubled in the older group, with 332/6153 (5.2%) variants observed among patients ≥40 years versus 12/482 (2.5%) in patients <40 years (Fig. 4a, c). Further, some variants, such as the highly transforming Q61K[34], were only observed in patients ≥40 years, with a linear increase correlating with increasing age (Fig. 4c).

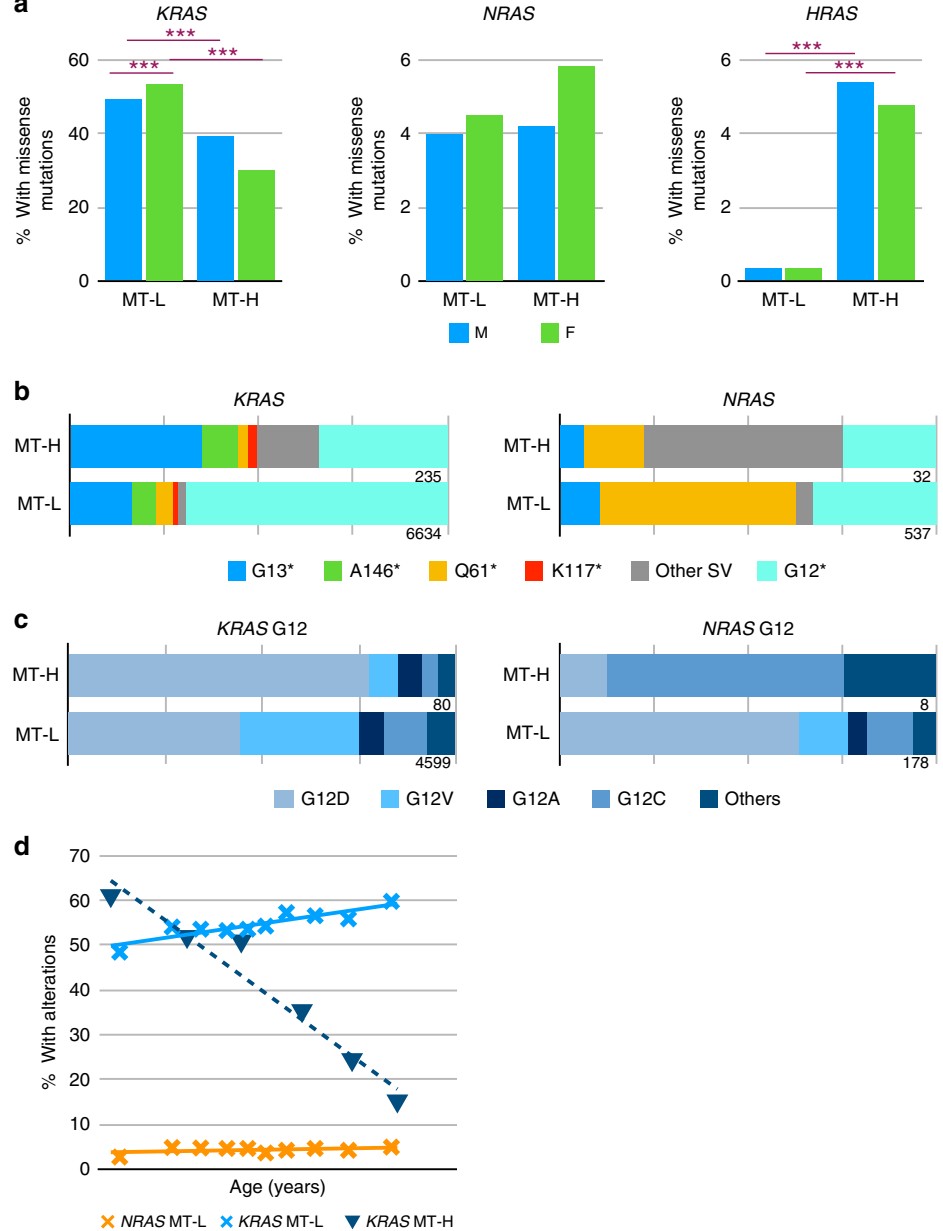

**Fig. 3** Codon missense mutational profile in *KRAS*-, *NRAS*-, and *HRAS*-based MSI-H/low TMB (MT-L) versus MSI-H/high TMB (MT-H) status. **a** Frequency of missense mutations in *KRAS*, *NRAS*, or *HRAS* as a factor of MT-L and MT-H status and gender in colorectal cancer (CRC) patients. ***$p < 0.005$. **b** Frequency of missense mutations affecting indicated residues in *KRAS* and *NRAS*, based on MT-H versus MT-L status. **c** Frequency of specific missense substitution in the G12 residue of *KRAS* and *NRAS*, based on MT-H versus MT-L status. **d** Frequency of missense mutations analyzed by age as a continuous variable in *KRAS* (in MT-L versus MT-H tumors) and *NRAS* (in MT-L tumors)

Similar numeric trends in mutational spectrum were observed in the MT-H population, including a greater proportion of G12 and a lesser proportion of A146 and Q61 mutations in the <40 years subgroup, and a greater diversity of mutations in older patients. However, these findings did not reach statistical significance due to lower overall numbers, with only 44/235 (18.5%) of MT-H patients <40 years (Supplementary Fig. 4c); further, trends based on analysis of age as continuous variable were not conclusive (Supplementary Fig. 4d).

Based on simple dichotomization (Fig. 4a, right), or analysis with age as a continuous variable (Fig. 4b, right), the *NRAS* mutational spectra did not differ significantly by age in the MSS patients. However, analysis using age as continuous variable suggested a non-linear trend towards a greater proportion of Q61

mutations and a lower proportion of G13 mutants with increasing age, except for the oldest patients (Fig. 4c, right). More data are needed to confirm the validity of these trends.

**Missense mutation variation dependent on colorectal subsite.** Among the 6634 defined missense mutations in *KRAS* in MT-L tumors, 983 (~15%) were from tumors originating in the rectum and 5651 (~85%) in the colon. For each CRC subsite, ~50% of the analyzed tumors had *KRAS* missense mutations. For the MT-H tumors, the corresponding numbers were 24 (10%) that arose in the rectum and 222 (90%) in the colon. The prevalence of *KRAS* missense mutations was much higher in patients with rectal cancer (21/35, ~60%) than in those with colon cancer (197/590,

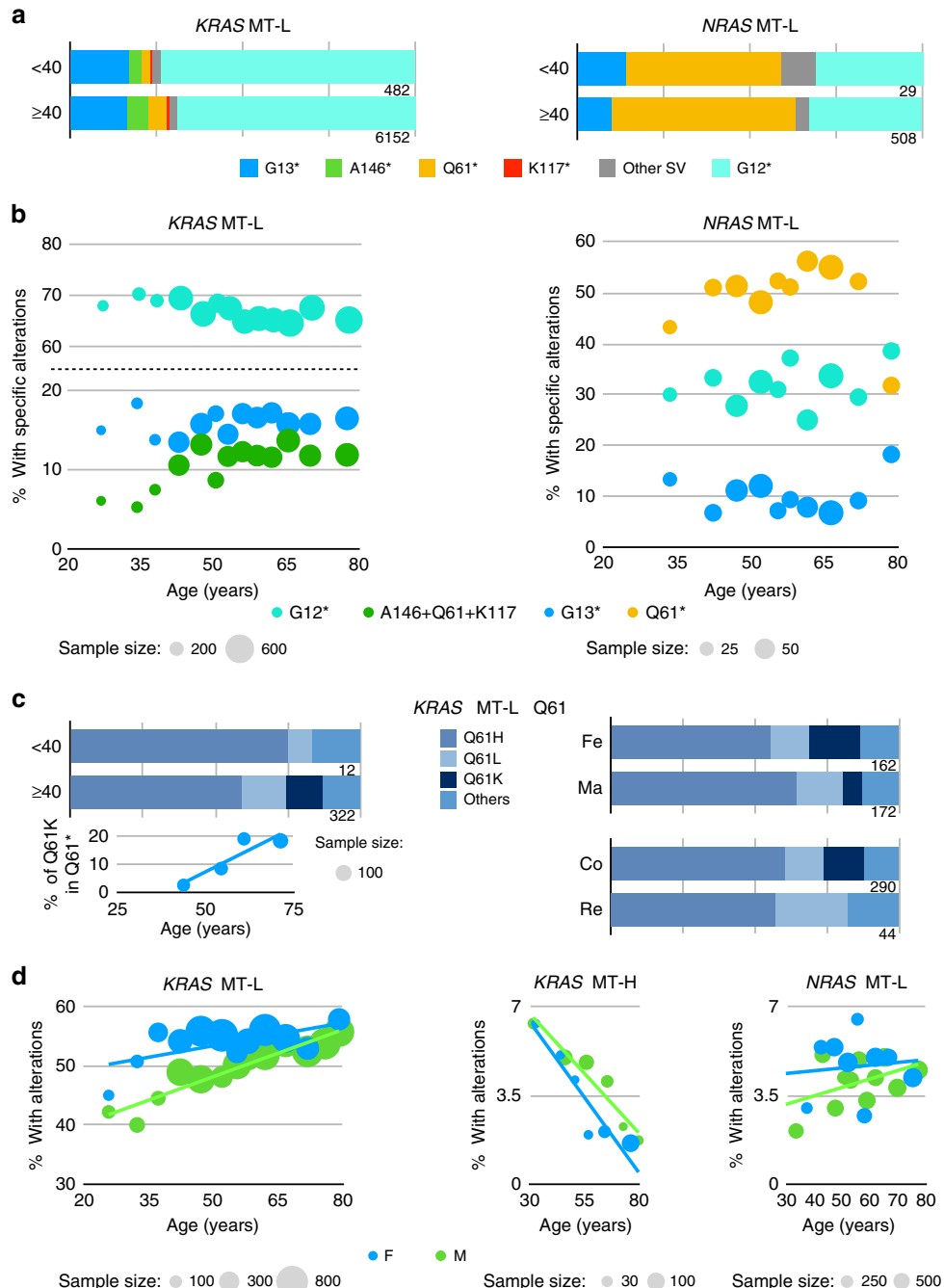

**Fig. 4** Codon missense mutational profile in *KRAS*, *NRAS*, and *HRAS* analyzed by age, gender, and colon versus rectal subsite. **a** Frequency of mutations in the indicated codons in *KRAS* and *NRAS* dichotomized based on age (<40 or ≥40 years) in MSI-H/low TMB (MT-L) colorectal cancer (CRC) patients. *Cumulative missense mutations. **b** Frequency of mutations in the indicated codons in *KRAS* and *NRAS* as a factor of age as a continuous variable in MT-L CRC patients. Lower abundance missense mutations affecting A146, Q61, and K117, are analyzed as a group. **c** Frequency of specific missense mutations *KRAS* Q61 based on age (<40 or ≥40 years), gender, and tumor subsite (colon, CO, versus rectum, RE) in MT-L CRC patients. **d** Frequency of missense mutations considered as a factor of age as a continuous variable and gender in *KRAS* (in MT-L versus MSI-H/high TMB (MT-H) tumors) and *NRAS* (in MT-L tumors)

~33%) ($p = 2.9^{E-03}$, Supplementary Fig. 4e). More specific information about the anatomical subsite of tumors within the colon or rectum was not available, and sample size did not permit detailed analysis of mutation patterns in colon versus rectum for MT-H tumors. The overall *KRAS* and *NRAS* spectra for the MT-L subgroup did not differ between the colon and rectum subsites (Supplementary Fig. 4f, g). However, one specific codon substitution was observed more commonly in colon tumors than in rectal tumors: Q61K, found in 40 MT-L colon tumors and 4

MT-H colon tumors, but not in rectal tumors ($p = 2.8^{E-03}$ for combined MT-L and MT-H groups, and $p = 4.6^{E-03}$ for MT-L tumors) (Fig. 4c, right).

**CRC *RAS* missense mutation variation dependent on gender.** Male patients with MT-L CRC had a slightly reduced frequency of *KRAS* mutations relative to female patients (49.4% and 53.5%, respectively; $p = 2.7^{E-05}$, Fig. 3a). Interestingly, age trends in the

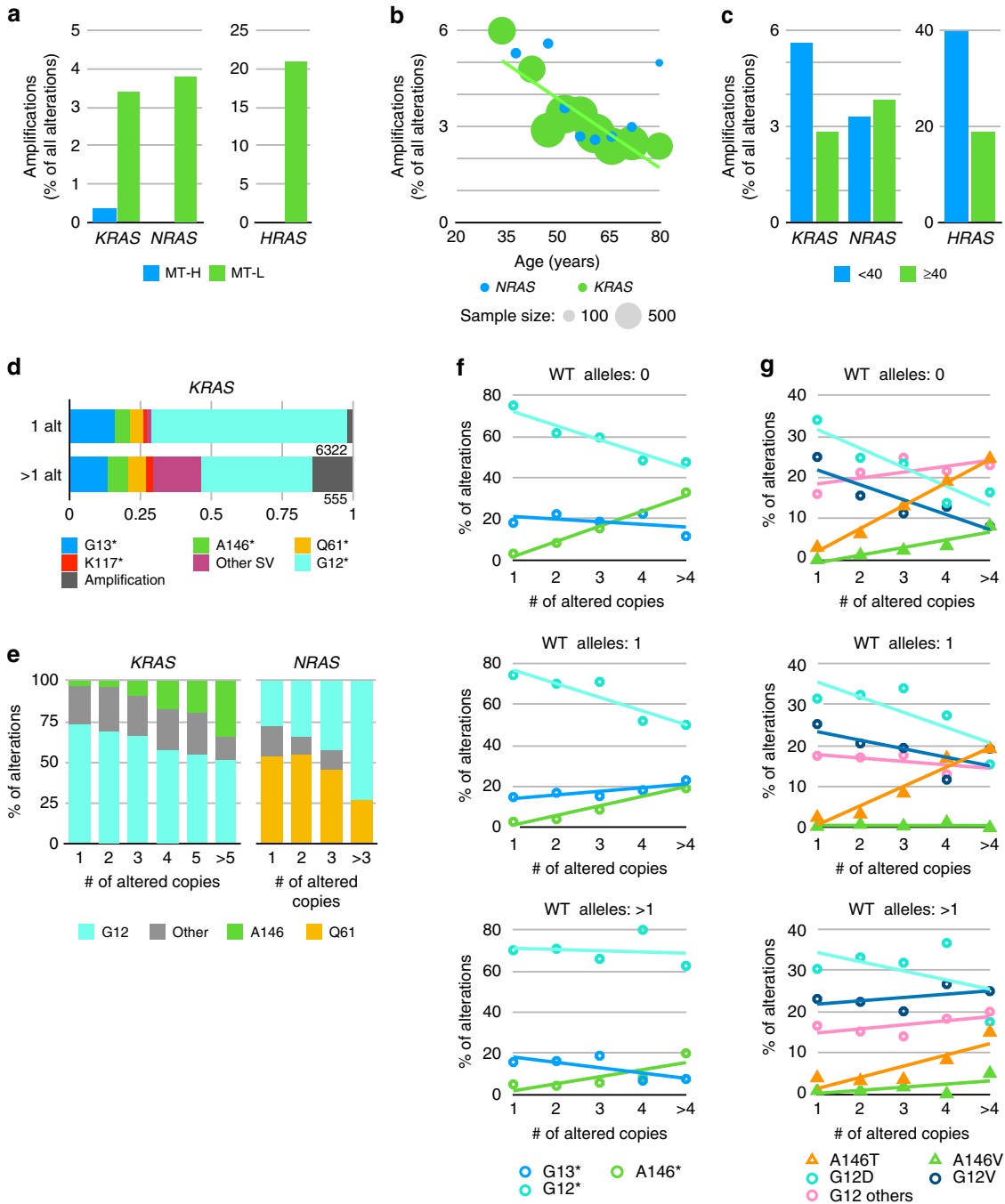

**Fig. 5** Copy number variation (CNV) and missense profile. **a** Frequency of amplification of the wt allele of *KRAS*, *NRAS*, or *HRAS* in the analyzed colorectal cancer (CRC) cohort, based on MSI-H/low TMB (MT-L) or MSI-H/high TMB (MT-H) status. **b**, **c** Frequency of amplification of the wt allele of *KRAS* or *NRAS* in MT-L CRC tumors, based on age as a continuous variable (**b**) or dichotomized as <40 or ≥40 (**c**). **d**. Frequency of amplification of mutated allele of *KRAS* is elevated in CRC tumors carrying one or more alteration (alt). *Cumulative observed missense changes. **e** Frequency of missense mutations in indicated codons as a factor of copy number in the mutated allele of *KRAS* or *NRAS*. **f** Frequency of missense mutations in indicated codons as a factor of increased copy number in the mutated allele of *KRAS*, in the context of indicated CNV in the wt allele of *KRAS*. *Cumulative observed missense changes. **g** Frequency of specific missense mutations in codons G12 and A146 as a factor of increased copy number in the mutated allele of *KRAS*, in the context of indicated CNV in the wt allele of *KRAS*. G12 others refers to combined rare missense mutations in G12 (e.g., G12S)

prevalence of *KRAS* missense mutations were distinct between males and females (Fig. 4d, left); an increase in mutational frequency with age was highly significant for males ($p = 3.8^{E-5}$), but not for females. Among MT-H cases, the total prevalence of *KRAS* mutations was higher in males (39.2% versus 30% in females, $p = 0.019$, Fig. 3a). The declining rate of such mutations

with age did not reach statistical significance for males ($p = 0.036$), but did for females (Fig. 4d, center, $p = 7.3^{E-04}$). Among specific missense mutations, the overall spectrum was similar between males and females (Supplementary Fig. 4h), with the exception of variants in Q61, where Q61K mutations were more frequent in females ($p = 1.3^{E-03}$) (Fig. 4c).

In contrast to the pattern observed with *KRAS* mutations, the frequency of *NRAS* SV mutations in MT-H tumors did not differ significantly from that in MT-L tumors ($p = 0.08$), likely because of the lower number of mutations (Fig. 3a). Similarly, there was no conclusive evidence for gender-specific age trends for the prevalence of *NRAS* missense mutations (Fig. 4d, right), and specific missense amino acid substitutions within *NRAS* did not show gender-specific variance (Supplementary Fig. 4i). For *HRAS*, no gender-related difference in the prevalence of missense mutations was detected (Fig. 3a); there were not enough cases of patients with these mutations to explore gender-specific age trends or gender-specific mutation spectra.

**Specific *RAS* mutation correlation with CNV.** Although most studies of *RAS* activation in cancer focus on the activating role of point mutations, allelic imbalance arising from either the amplification of the mutated allele or loss of the wt allele can affect tumorigenicity and therapeutic outcome[35,36]. Among all CRC tumors, 225 amplifications (defined as six or more copies) of wt *KRAS*, 21 of wt *NRAS*, and 11 of *HRAS* were observed among 12,711 MT-L tumors, while only one amplification of *KRAS* and none of *NRAS or HRAS* were detected in 625 MT-H tumors. Thus, amplifications of wt *RAS* genes are much more prevalent in MT-L tumors ($p = 3.7^{\mathrm{E}-04}$ for *KRAS*, and $p = 8.3^{\mathrm{E}-05}$ for all *RAS* genes combined) and comprise a much higher fraction of total *RAS* gene alterations in MT-L tumors (Fig. 5a), although the numbers did not reach statistical significance except for *HRAS* ($p = 8.4^{\mathrm{E}-04}$).

We analyzed *RAS* amplification for association with age or tumor subsite to the extent sample size allowed significant conclusions. There was no difference in the frequency of *KRAS* amplifications between the colon and rectal subsets, overall. However, greater prevalence of *KRAS* amplification in CRC patients of younger age achieved borderline statistical significance ($p = 5.9^{\mathrm{E}-03}$, Fig. 5b). Dichotomization by age 40 years revealed that of 515 *KRAS* alterations in MT-L tumors from younger patients, 31 were amplifications (5.6%); from 6411 *KRAS* alterations in patients ≥40, 194 were amplifications in *KRAS* (2.8%), representing a 2-fold lower level in older patients ($p = 6.8^{\mathrm{E}-04}$) (Fig. 5c). Interestingly, *KRAS* amplifications also relatively frequently co-occurred with a *RAS* SV mutation in the same patient. Among 6322 MT-L CRCs with a single *KRAS* alteration (either amplification or SV), only 2.2% bear a *KRAS* amplification (140 occurrences). In contrast, in MT-L CRCs bearing two or more *KRAS* alterations, *KRAS* amplifications represent 83 out of 555 alterations (15%)—almost a 7-fold increase ($p < 2.2^{\mathrm{E}-16}$) (Fig. 5d).

For *NRAS* amplifications, no difference was found using dichotomization by age 40 years (Fig. 5c). However, comparative analysis of amplification rate in patients diagnosed at different age groups suggests that the rate of amplification may decline in tumors arising in older patients (Fig. 5b). The number of samples with *HRAS* amplifications was too low for detailed dissection of age–amplification relationships, but age 40 dichotomization also suggests an elevated prevalence in younger patients will emerge in analysis of a larger cohort (Fig. 5c).

Analysis of the profile of common *KRAS* and *NRAS* mutations in MT-L tumors in the context of allelic variation revealed some striking differences between distinct codon variants (Fig. 5e–g). Comparison of the pattern of missense mutations to the patterns of allelic gain and loss revealed specific correlations involving some of the more common missense mutations (Fig. 5e). Mutations affecting *KRAS* G12, present at 69.3% overall among missense mutations, were much less likely to be found when increasing copies of the mutant allele were present (representing

72.6% of all SV mutations found as one copy, but only 50.9% of the pool of *KRAS* SV mutations with five or more altered copies; association between copy number and G12 mutation fraction $p = 1.73^{\mathrm{E}-11}$). The opposite pattern was observed with mutations in *KRAS* A146, representing 5.8% of overall mutations; 3.6% of samples with a single SV showed a mutation in A146, but 34.5% in samples with five or more mutated copies ($p < 2^{\mathrm{E}-16}$). In contrast, other missense variants, such as those affecting G13, do not have similar variance as a factor of copy number (Fig. 5e). A change in frequency of specific mutations based on amplification of the mutated allele was also observed with *NRAS*; surprisingly, here G12 mutations were more common in the context of higher copy number, whereas Q61 became less common (Fig. 5e).

Among the more common missense mutations, trends of association with CNV of the mutated allele were most strongly detected for missense variants of G12 and A146. In the setting of 0 copies of wt *KRAS*, G12 missense mutations were less likely to be amplified, and A146 more likely (Fig. 5f). As the number of wt copies of *KRAS* increased, these trends were lost, reflecting a significant relationship between wt copy number and specific mutations ($p = 1.5^{\mathrm{E}-05}$ for G12 and $3.8^{\mathrm{E}-04}$ for A146). Among specific missense substitutions, G12D and G12V showed strong copy number dependence, but not G12A, C, or S; A146T showed a stronger copy number effect than the less common A146V (Fig. 5g). These patterns are also more prominent in tumors where the wt *KRAS* allele has been lost (Fig. 5g).

## Discussion

Metastatic CRC serves as an example of the importance of molecular characterization in defining the optimal treatment paradigm for an individual patient. A limited number of targeted therapies have been approved for CRC, with anti-EGFR antibodies such as cetuximab and panitumumab the most successful to date. *KRAS* and *NRAS* are central downstream effectors of EGFR. Two landmark studies demonstrated improved OS and progression-free survival in patients with wt *KRAS* treated with anti-EGFR therapy, while showing no benefit to those with *KRAS* mutations[37,38]. While initial studies investigated only mutations in codons 12 and 13 of the *KRAS* gene, additional studies have demonstrated other mutations that predict for resistance to EGFR-targeted treatment, those that do not predict for resistance, and the predictive value of expanded *RAS* testing, including *NRAS* and *BRAF*[37,39,40]. Hence, there is considerable value in establishing the overall profile of *RAS* mutations relevant to CRC.

The large sample size in this study reinforces recent findings of other large studies, and also allows some statistically significant conclusions. As one point of interest, the rate of hypermutated patients observed in this study at 4.5% is significantly lower than that reported by the TCGA and others at ~13–15%[24]. This difference reflects the fact that the patients analyzed by FM all presented with metastatic disease. MSI-H CRC has been reported to be less frequently metastatic than MSS CRC, with only 4–5% of stage 4 patients with MSI-H;[41] the frequencies reported here are more likely to be relevant to this advanced population. Others have reported that MSI-H is more common in elderly females due to higher prevalence of CpG island methylator phenotype (CIMP) and MLH1 hypermethylation;[42,43] this study did not identify such a correlation. This difference, also, may reflect differences in the study pool.

Others have found that *KRAS* mutant tumors were less likely than *KRAS* wt tumors to be deficient in mismatch repair pathways (7% versus 14% $p < 0.001$)[44], agreeing with the findings of the present study. The data presented here both confirms previously identified mutational hotspots reported for CRC, or for other cancers, and also identifies some new hotspots, including

V9, V14, A68, R164, K176, and K180. Some of these (e.g., V14[45]) have previously been reported in sporadic tumors or in RASo-pathies, which include Noonan syndrome, Costello syndrome, cardiofaciocutaneous syndrome, and neurofibromatosis type 1[16,45], and are typically associated with a milder RAS-activating function. Mutations that lead to these RASopathies impact intrinsic and GEF-induced nucleotide exchange, or affect intrinsic and/or GAP-induced hydrolysis. Modeling of these mutations on the 3D structure of KRAS places these residues adjacent to known critical functional areas for these processes (Fig. 2). Two pre-viously unreported hotspot mutations, K176 and K180, target the positively charged lysine tract that characterizes the KRAS-4B isoform; interestingly, one study has indicated that these residues interact with high affinity with the GDP-bound Switch1/Switch2 region, leading to the proposal they support auto-inhibition of KRAS[46], which would provide one potential mechanism for activating function. Another interesting feature of the study is the recognition that specific common mutations are associated in characteristic ways with CNV of either the mutated or wild-type allele, expanding on earlier studies in this area using smaller cohorts[17,35].

There is emerging awareness that the clinical behavior of colon cancer is impacted by the specific site of origin within the colon: for example, with tumors on the right side are associated with reduced survival relative to left-sided CRCs, data from multiple clinical trials demonstrating that left-sided colon cancer patients are more responsive to anti-EGFR therapy than right-sided cancers[47], and an ongoing investigation into whether different mutational spectra exist in left versus right colon[44,48,49]. KRAS mutations, for example, have been reported to be more common in proximal and cecal tumors, associated with specific morpho-logical features[50,51]. While the dataset in this study does not discriminate left from right colon, we did find notable differences in rectal cancers, as compared to colon cancers, including dif-fering mutational spectra.

The age-related differences identified here are intriguing. Examination of young patients with CRC in small studies has shown a greater proportion of CIMP-low tumors in the sporadic (non-Lynch syndrome) cohort[52]. The Wnt/β-catenin pathway has also been identified as more commonly activated in young patients[53]. To date, examination of the RAS pathway in these patients has been limited, and has been inconclusive, with a wide range of rates of KRAS mutations reported in young patients (4–54%)[54,55], and no prior investigations into specific mutations are known. Finally, some of the data in this study hints at specific associations connecting age, gender, and tumor subsite with specific mutations. Of particular interest, the Q61K mutation is associated with patients who are older at testing, female, and diagnosed with colon cancer; this intriguing set of relationships suggests that investigation of cross-category linkages may identify unique disease subtypes that bear specific scrutiny. While data regarding specific indication for sequencing is unavailable for the samples in this study, the size of the dataset and varied prove-nance of the samples decreases the risk of the findings being due to selection bias. Finally, the clinical implications of these data would be illuminated with linked survival data, but these are not available for analysis. Nevertheless, this study and linked dataset should serve as a useful benchmark for future investigations.

## Methods

**Comprehensive genomic profiling**. CGP was performed using the Foundatio-nOne assay (FM Inc., Cambridge, MA, USA), as previously described in detail[56]. Briefly, the pathologic diagnosis of each case was confirmed by review of hema-toxylin- and eosin-stained slides and all samples that advanced to DNA extraction contained a minimum of 20% tumor cells. Hybridization capture of exonic regions from 315 cancer-related genes was applied to ≥50 ng of DNA extracted from formalin-fixed, paraffin-embedded clinical cancer specimens. These libraries were sequenced to high, uniform median coverage (>500×) and assessed for base sub-stitutions, short insertions and deletions, copy number alterations, and gene fusions/rearrangements. MSI microsatellite track from the UCSC genome browser was aligned with short variants detected in KRAS and NRAS to determine whether specific mutations were associated with microsatellites. Comparison data sets for studies with information on KRAS and NRAS mutation status, and gender, age, and tumor subsite were collected from the cBioPortal database (http://www.cbioportal.org/index.do).

**Statistical assessments of data**. The relationships between mutations and patient characteristics were assessed using Fisher's exact tests (including determination of significant difference between the mutational spectra of dichotomized age, gender, or subsite groups) and multivariable logistic regression models. For Fig. 4b, d and 5b, we determined whether age was associated with probability of mutations of interest using Pearson's correlation, weighted to account for uneven sample sizes within age groups. To allow for multiple mutations within a patient, the predictors in these models were binary indicators for the presence/absence of particular mutations of interest. Separate models were conducted for the genes (KRAS/NRAS) and patient characteristics (gender, younger age, tumor site, MSI). Overall sig-nificance was assessed using a likelihood ratio test. Conditional inference trees[57] were also used to determine which mutations were most strongly associated with patient characteristics of interest. CNV analyses were conducted using logistic regression analyses in the subgroup of patients with a KRAS mutation. Analyses were done overall and stratified by the number of KRAS wt CNVs. To determine if a frequently mutated site on the RAS proteins constitutes a mutational hotspot, we have used a binomial distribution model with a cumulative probability cutoff of 0.005.

**Visualization of folded domains and mutation spectra**. Figures were prepared using the program PyMOL[58]. The KRAS structure used is the GDP-bound full-length KRAS4b (PDB code: 5TAR)[59]. The NRAS structure used is the GppNHp-bound NRAS structure, which is 166 residues (PDB code: 5UHV)[32]. The HVR is not represented in this resolved structure. Another available NRAS structure (PDB code: 3CON) was not used as it lacks information on residues 61–71. The HRAS structure[32,60] (PDB code: 1CTQ) used is the GppNHp-bound HRAS structure, which is 166 residues; this structure does not resolve the HVR.

**Reporting summary**. Further information on research design is available in the Nature Research Reporting Summary linked to this article.

## Data availability

The data that support the findings of this study appear in Supplementary Data 1. Clinical data for this cohort also appear in Supplementary Data 1. Because of the restrictions on disclosure based on the Health Insurance Portability and Accountability Act Privacy Rule and the Genetic Information and Nondiscrimination Act, we are unable to provide access to the raw sequencing reads because they include germline information. Furthermore, we provide qualified researchers access to additional detailed data through application to Foundation Medicine (https://www.foundationmedicine.com/insights-and-trials/foundation-insights).

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

## Acknowledgements

We thank J. Karanicolas and R. Dunbrack for consultation on analysis of protein structure. The authors were supported by NCI Core Grant P30 CA006927 (to Fox Chase Cancer Center), NIH R01 DK108195 (to E.A.G.), NIH R01 CA229259-01 (to C.L.), by a subsidy of the Russian Government to support the Program of Competitive Growth of Kazan Federal University (to I.G.S.), and Department of Defense Career Development Award W81XWH-18-1-0148 (to S.A.), and by the Colorectal Cancer Alliance (to E.A.G., J.E.M.).

## Author contributions

C.C., G.F., M.C., V.M., S.Ali. and J.S.R. collected data. I.G.S, E.H. S.Arora., and J.S.R. performed data analysis. E.A.G., I.G.S. S.Arora., C.L. and J.E.M. designed the study and wrote the manuscript.

## Additional information

**Competing interests:** C.C., G.F., M.C., V.M., S.Ali, and J.S.R. are employed by FM Inc., and own stock in FM Inc. The remaining authors declare no competing interests.

