## [Peer Review File · Nature Communications]

Reviewers' comments:

Reviewer #1 (Remarks to the Author):

The current study is likely the largest analysis about RAS mutational profile. The authors identified new mutation hotspots and mutation spectrum in KRAS and NRAS based on the age of diagnosis, MSI status, and colon versus rectum subsite. The data are very well presented and appears strong with a large number of patients. The authors also discussed the current knowledge, rationale, and available data.

The authors should put "p." for all amino acid position numbers in all places including the text, tables, and figures. E.g., p.Q61 (not Q61), p.G12, p.G13 for amino acids, and p.G12D, p.G13D, etc. for mutations.

The authors should describe nucleotide changes for all mutations along with amino acid changes in parenthesis. E.g., c.35G>A (p.G12D). The authors sequenced DNA but not amino acids. Different nucleotide changes can result in the same amino acid changes. E.g., both c.35G>A and c.35_36delinsAC result in p.G12D. Thus, specific nucleotide changes should be described.

Were MSI-H tumors more prevalent in the elderly female population? Overall prevalence of MSI-H was low in this study. Because previous studies have shown that MSI-H is more common in elderly females due to higher prevalence of CIMP and MLH1 hypermethylation.

It is likely that very young patients and old patients were referred (and tested) in different ways, which led to differential selection bias. How can the authors address this problem?

The authors have shown that RAS mutational spectrum differences between colon and rectum. Previous studies have shown KRAS mutations are the most prevalent in cecal cancers compared to cancers in other colorectal sites (Yamauchi et al. Gut 2012; Rosty et al. Mod Pathol 2013). Many studies showed higher prevalence of KRAS mutations in proximal colon cancers (that contained cecal cancers of course) than in distal colorectal cancers. Can the authors examine RAS mutations in cecal cancers? If not, at least the authors should discuss the link between KRAS mutations and cecal cancers.

With this large number of cases, the authors should use the stringent alpha level of 0.005 recommended by Benjamin et al. Nat Hum Behav 2018, unless it is adjusted for multiple testing by another method. In page 6, eg, the authors used 0.05 as the alpha level, which can easily lead to false positive findings.

The authors have shown that p.G12 mutations, p.A146 mutations and other SV in KRAS predominated MSI-H tumors. Are these other mutations in MSI-H tumors associated with microsatellite region (repeat nucleotide sequencing)?

HRAS mutations are rare. It would be nice to provide data on HRAS mutations if any.

If survival data are available, survival analysis with novel hotspots would be interesting. I understand if such data are not available.

Reviewer #2 (Remarks to the Author):

The study offered a much needed comparison of the genomic landscapes of colon and rectal cancers, with a deeper analysis on the mutations of KRAS and NRAS. A larger dataset provided the statistical power to recapitulate a tendency for younger patients to have MSI-H tumors in both colon and rectal cancers, though an overall higher rate of MSI-H in colon cancers. Further, several often mutated codons in KRAS were identified in regions of the protein spatially close to other known oncogenic hotspots, including codons 12 and 13. Inspection of the mutations of KRAS in relation to CNV revealed a simultaneous increase in the frequency of A146 mutations and decrease in the frequency of G12 mutations; the same pattern holds for G12 and Q61 mutations in NRAS. Finally, a potential link between the CNV of the wt KRAS allele and frequency of specific KRAS mutations was noted.

The authors highlighted several key trends of KRAS mutations in CRC that warrant further exploration, however, the study, while being difficult to comprehensively assess due to vague descriptions of statistical analyses, also lacked any insightful mechanistic conclusions on the nature of KRAS in colon and rectal cancers.

Major Points

1. The authors indicate in Table 1 that 4.53% of their CRC samples were MSI-H, while other sources, namely the TCGA initiative, have repeatedly found the fraction of hypermutants to be between 15-20% of CRC tumors. This discrepancy is difficult to account for since the data presented here is from selectively sequencing 315 genes while the TCGA employed whole exome or genome sequencing (WE/GS). Further, it is unclear why most samples had prior information on their MSS/MSI-H categorization and TMB values, 3,386 had only TMB data, and 384 samples had neither. The authors should address where this prior information came from and why it is absent in a substantial number of samples. Finally, the MSS/MSI-H status of the samples with only TMB data were classified by either having fewer or at least 16 mutations. Presumably, this low of a threshold was a consequence of the sequencing technology used. It would be instructive if the authors were to apply the same cutoff to samples of WE/GS by extracting the genes used by the FMI sequencing system and applying the same threshold for TMB. I find the categorization of hypermutants from WE/GS to be more convincing and would be more confident in the results of the paper if they are able to correctly classify perviously identified hypermutants with the technique reported in the paper. This is an important point for the authors to clarify because they note an increase in MSI-H tumors from patients <40 years old, though the rate they report is ~15%, the normally cited frequency for the general population.

2. While I understand the authors' purpose for applying a threshold of 40 years-of-age between "young" and "old," I'm not sure they provide a reason for instituting a threshold at all. With their large sample size, I would rather the authors treat age as a numeric variable instead of coercing it to a nominal variable. Additionally, the authors state that, "Age 40 has been used to examine the increasing incidence of colorectal cancer in young people, due to the lack of colorectal cancer screening below age 40, thereby removing screening as a source of bias," [lines 103-105]. Could this influence the results towards a bias for patients <40 years old to have MSI-H? In other words, it is possible that MSI-H tumors are more likely than MSS tumors to be discovered without screening, artificially boosting the frequency of <40 year olds having MSI-H tumors. Again, treating age as a numerical variable could assuage these concerns.

3. The authors should state how they determined if a frequently mutated site on KRAS constituted as a "mutational hotspot." From Figure 2A, it seems that a site was deemed a hotspot if it appeared ≥ 10 times. In a population of 7,109 samples, this means the authors define a hotspot as a mutational frequency of ~0.14%, a threshold I am uncertain the rest of the field will agree with. Plotting the mutation burden instead of the frequency (as done in the rest of the paper) may be misleading. This is only magnified by displaying the y-axis on a logarithmic scale. If mutational frequency were plotted on a linear scale, codons mutated 10 times in a population of 7,109 would not likely be classified as high-frequency events.

4. It would be beneficial for the authors to more explicitly state which statistical tests they used for each figure. Further, the use of conditional inference trees or other more sophisticated and less common tools should be specifically justified over those simpler and more common. Additionally, the authors did not explain clearly why or when they used the Fisher exact test, z-ratio test, or the chi-squared test when they did. This was a confusing point as they are all tests for similar questions (are the proportions of a nominal variable different between two populations). Generally, it seems that the Fisher exact test is the most highly recommended. Finally, the figure legends do not explain the error bars in Figure 1C-E nor are any of the “ * ” symbols used to show statistical significance. It is not intuitive as to what they are describing and should be explicitly stated in the respective figure legends.

5. Do the authors adjust for multiple hypothesis testing throughout the manuscript? This is important to include in many of the figures to account for false positives. For example, in Figure 1C, there are four hypotheses tested: M–M, F–F, $F < 40 - M < 40$, and $F \geq 40 - M \geq 40$.

6. The authors should state how the lines in Figure 4E and F were constructed. Specifically, why are some non-linear where there does not appear to be a clear curvilinear relationship. This is important to prevent future readers from mis-interpreting the graphs.

Minor Points

1. A brief description of the technology used to gather the data at the heart of this study included at the beginning of the results section would go a long way in framing the data for the reader.

2. Please include a comment on the nature and handling of samples from female or male discordant patients distinguished in Table 1 (e.g. are they included in figures comparing across sexes?).

3. In Figure 1A, is “FM” referring to “FMI” used elsewhere in the manuscript? Also, the legend “PAD, publicly available datasets” is covering part of the chart.

4. A higher-dimensional chart showing the information contained in Figure 1C-E may allow for easier comparison between all of the factors at hand.

5. There are a few instances of numbers without comma-separation every 3 digits to the left of the decimal point, while most do include the comma (eg. “1267 CRC tumors” [line 117] versus “~25,000 tumors of all types” [line 118]. Commas should be included as per Nature’s formatting standards.

6. There are inconsistencies between the capitalization and italicization of KRAS, NRAS, and RAS. This should be standardized based on whether the use is in reference to the gene or protein.

7. On line 133, “Supplemental” is missing from “(Figure S2B, C).”

8. Figure 2A would benefit from a line showing the authors’ threshold for “hotspot” classification (along with the points made in #3 of Major Points, above).

9. Figure 2F is missing the total number of tumor samples for MSI-H and MSS. This was a great addition in many of the other plots of similar format throughout the paper (including the adjacent panel G).

10. By eye, there appear to be interesting correlations between Figure 3C and F and between Figure 3B and E. Seeing as the age of the patient and location of the tumor do not seem to be independent (Figure 1E), the graphing of location, age, and mutation of the codon (one chart for 12 and another for 61) could show some interesting associations.

11. Figure 3 B-F, H appear to be stretched or compressed, especially compared to panel A (this is an issue in other figures, too, though most noticeably here). Also, some of the numbers indicating sample size are out of place in the same panels.

12. Is line 222 intended to say “KRAS” instead of “RAS”? Otherwise, the statement may need clarification.

13. Some P values were not converted to scientific notation in cases where they should be (including on lines 225 and 244).

14. In my opinion, Figure 4E and F would be improved if reconfigured to adhere to more standard labeling practices. The label along the y-axes does not indicate what is measured. There are no labels

along the x-axes. In addition, It does not appear that G12S, though included in the legend of panel F, is plotted in any of the graphs. Lastly, why are some of the lines dashed and others not in panel F? Is this intended to represent a property of the data? If so, it requires clarification in the legend.

15. There are two different spellings of "RASopathies" between lines 269 and 271.

16. Please include a legend with the chart in Supplemental Figure 1. Also, if the authors continue to use the threshold of 16 mutations between MSS and MSI-H, it would be helpful to include a vertical line indicating this point on the chart.

Response to Reviewers

Reviewer #1:

The current study is likely the largest analysis about RAS mutational profile. The authors identified new mutation hotspots and mutation spectrum in KRAS and NRAS based on the age of diagnosis, MSI status, and colon versus rectum subsite. The data are very well presented and appears strong with a large number of patients. The authors also discussed the current knowledge, rationale, and available data.

Point 1. The authors should put “p.” for all amino acid position numbers in all places including the text, tables, and figures. E.g., p.Q61 (not Q61), p.G12, p.G13 for amino acids, and p.G12D, p.G13D, etc. for mutations....The authors should describe nucleotide changes for all mutations along with amino acid changes in parenthesis. E.g., c.35G>A (p.G12D).The authors sequenced DNA but not amino acids. Different nucleotide changes can result in the same amino acid changes. E.g., both c.35G>A and c.35_36delinsAC result in p.G12D. Thus, specific nucleotide changes should be described.

Response. We agree with the reviewer that the convention of p.Q61 (versus Q61) is more standard for articles directed towards the genetics community; however, the Q61 formulation is more typical for articles read by the cancer signaling community. A study of RAS family members is clearly of interest to both audiences. Also, as a pragmatic note, some of the main figures are already extremely visually “busy”, and extending the name of the proteins would make them more so. However, we do recognize that it is important to provide relevant mutational numbers for those members of the cancer community interested in the frequency of distinct nucleotide changes found in the various subsets of KRAS, NRAS, and (now) HRAS analyzed here. To reconcile these considerations, we have left the nomenclature as originally provided in the main text and figures. However, we have now provided a new Supplementary Table S1 in which we provide a detailed breakdown of the significant majority (>97%) of the observed nucleotide changes and associated protein changes, with the notation as requested, for interested researchers. We have information regarding the remaining ~ 3% of very rare changes, and will make these available to interested readers on request, as noted in the Figure Legend; we have excluded them for now, as they make the supplemental Table unwieldy.

Point 2. Were MSI-H tumors more prevalent in the elderly female population? Overall prevalence of MSI-H was low in this study. Because previous studies have shown that MSI-H is more common in elderly females due to higher prevalence of CIMP and MLH1 hypermethylation.

Response. We believe the prior studies referred to include PMID: 12627505, by Kakar et al (2003), and PMID: 18165277 by Ogino and Goel (2008). As shown in the graphs presented in a Fig.1C, the fraction of tumors designated as MT-H (MSI-H/TMB-H) was not increased in the elderly female population. We have also performed similar analysis for colorectal tumor both using MSI status alone, or on the basis of high TMB (new Supplementary Fig.2), and found no elevation in this sex/age group. We have added a citation to the earlier studies, and now mention this difference in the revised discussion section.

Point 3. It is likely that very young patients and old patients were referred (and tested) in different ways, which led to differential selection bias. How can the authors address this problem?

Response. Testing by Foundation Medicine and other genomic profiling sources, including academic institutions, is currently not performed based on standardized criteria. Institutional and physician preference play large roles, and the rationale in the case of each individual patient is typically not recorded in the records associated with sequence data. Hence, based on 2019 standards, this is a virtually unanswerable question for any large data set. The best response that we can make is that all the patients in the dataset were diagnosed with metastatic colorectal cancer at varying timepoints during their treatment. Therefore, though selection bias is a possibility, the dataset is a large representative sample set of the general population diagnosed at this disease stage. We have added commentary on these points to the Discussion.

Point 4. The authors have shown that RAS mutational spectrum differences between colon and rectum. Previous studies have shown KRAS mutations are the most prevalent in cecal cancers compared to cancers in other colorectal sites (Yamauchi et al. Gut 2012; Rosty et al. Mod Pathol 2013). Many studies showed higher prevalence of KRAS mutations in proximal colon cancers (that contained cecal cancers of course) than in distal colorectal cancers. Can the authors examine RAS mutations in cecal cancers? If not, at least the authors should discuss the link between KRAS mutations and cecal cancers.

Response. We are in complete agreement with the reviewer that it would have been highly desirable to include information about sidedness, and cecal origin of tumors. Unfortunately, sidedness data was unavailable in this dataset. So, though the question is extremely intriguing, it cannot be answered due to the lack of data. However, we have modified the Discussion to cite the papers noted, and to discuss these issues.

Point 5. With this large number of cases, the authors should use the stringent alpha level of 0.005 recommended by Benjamin et al. Nat Hum Behav 2018, unless it is adjusted for multiple testing by another method. In page 6, eg, the authors used 0.05 as the alpha level, which can easily lead to false positive findings.

Response. We agree that multiple testing is an important consideration (a point also noted by Reviewer 2). As suggested, we have therefore adopted the $\alpha=0.005$ threshold for declaring statistical significance. As a result, a few of the less important analyses that yielded significant differences based on the 0.05 criterion have lost significance; Figure panels reporting these data have been moved to Supplemental Figure S4. However, for the bulk of the work reported in this study, the conclusions hold.

Point 6. The authors have shown that p.G12 mutations, p.A146 mutations and other SV in KRAS predominated in MSI-H tumors. Are these other mutations in MSI-H tumors associated with microsatellite region (repeat nucleotide sequencing)?

Response. During data collection, we superimposed the MSI microsatellite track from the UCSC genome browser with short variants detected in KRAS and NRAS. There was only one intronic microsatellite locus (in KRAS), and it did not overlap with baited regions and short variants. Therefore, the other mutations are not associated with the microsatellite region. We have added brief text to Methods and Results (near discussion of Figure 2C) addressing this point.

Point 7. HRAS mutations are rare. It would be nice to provide data on HRAS mutations if any.

Response. This was an excellent idea; we have now included the data on HRAS mutations in our study, reflected in a number of modified figures. The analysis of these data allowed us to identify additional hotspots in HRAS, not previously reported in CRC (to the best of our knowledge). As noted by reviewer, HRAS mutations are rare (a total of 101 alterations, including 88 structural variations) were detected in the complete dataset); in spite of the fact this is the largest dataset reported for CRC, the scarcity limited our ability to identify significant trends in these mutations. However, intriguingly, these data do suggest differences in HRAS mutation based on MSI-H versus MSS status (higher in MSI-H), and in HRAS amplification (higher in young patients). These data have been added to the Figures, Results, and Discussion.

Point 8. If survival data are available, survival analysis with novel hotspots would be interesting. I understand if such data are not available.

Response. We entirely agree with the reviewer that this information would be highly desirable. Unfortunately, although we and other consortia are working towards generating such granular information, this is not yet the case. We have added a comment to this effect in the discussion.

Reviewer #2:

The study offered a much-needed comparison of the genomic landscapes of colon and rectal cancers, with a deeper analysis on the mutations of KRAS and NRAS. A larger dataset provided the statistical power to recapitulate a tendency for younger patients to have MSI-H tumors in both colon and rectal cancers, though an overall higher rate of MSI-H in colon cancers. Further, several often-mutated codons in KRAS were identified in regions of the protein spatially close to other known oncogenic hotspots, including codons 12 and 13. Inspection of the mutations of KRAS in relation to CNV revealed a simultaneous increase in the frequency of A146 mutations and decrease in the frequency of G12 mutations; the same pattern holds for G12 and Q61 mutations in NRAS. Finally, a potential link between the CNV of the wt KRAS allele and frequency of specific KRAS mutations was noted.

The authors highlighted several key trends of KRAS mutations in CRC that warrant further exploration, however, the study, while being difficult to comprehensively assess due to vague descriptions of statistical analyses, also lacked any insightful mechanistic conclusions on the nature of KRAS in colon and rectal cancers.

Major Points

Point 1. The authors indicate in Table 1 that 4.53% of their CRC samples were MSI-H, while other sources, namely the TCGA initiative, have repeatedly found the fraction of hypermutants to be between 15-20% of CRC tumors. This discrepancy is difficult to account for since the data presented here is from selectively sequencing 315 genes while the TCGA employed whole exome or genome sequencing (WE/GS). Further, it is unclear why most samples had prior information on their MSS/MSI-H categorization and TMB values, 3,386 had only TMB data, and 384 samples had neither. The authors should address where this prior information came from and why it is absent in a substantial number of samples. Finally, the MSS/MSI-H status of the samples with only TMB data were classified by either having fewer or at least 16 mutations. Presumably, this low of a threshold was a consequence of the sequencing technology used. It would be instructive if the authors were to apply the same cutoff to samples of WE/GS by extracting the genes used by the FMI sequencing system and applying the same threshold for

TMB. I find the categorization of hypermutants from WE/GS to be more convincing and would be more confident in the results of the paper if they are able to correctly classify previously identified hypermutants with the technique reported in the paper. This is an important point for the authors to clarify because they note an increase in MSI-H tumors from patients <40 years old, though the rate they report is ~15%, the normally cited frequency for the general population.

Response. We agree that the MSI-H numbers are discordant with the higher TCGA-reported numbers. However, our numbers are closer to those reported in several other recent studies. For example, the fraction of MSI-H tumors reported at the AACR from an analysis of ~3650 CRC patients profiled by Caris was 5.9 % (*Journal of Clinical Oncology* 34 number 15, suppl (May 2016) 3599-3599.. Most important, in the dataset we analyze here, all patients had metastatic, stage IV CRC. In the total population (all stages), the MSI-H rate is expected to be between 10-15%. However, in the stage IV setting, this number has been reported to be 4-5%, because MSI-H tumors have a reduced metastatic potential (Koopman et al, PMID: 19165197). Hence, we do not believe the discordance reflects defects in the current standard of clinical sequencing technology, but rather, differences in the patient pool; we also hypothesize that the original estimates may have been a bit high, and are being corrected as larger groups of patients are being analyzed. We agree with the reviewer that it is important to be clear about differences in the numbers reported here versus those in the literature, and we have modified the Discussion to include the Koopman reference and other points made in this response.

Point 2. While I understand the authors' purpose for applying a threshold of 40 years-of-age between "young" and "old," I'm not sure they provide a reason for instituting a threshold at all. With their large sample size, I would rather the authors treat age as a numeric variable instead of coercing it to a nominal variable. Additionally, the authors state that, "Age 40 has been used to examine the increasing incidence of colorectal cancer in young people, due to the lack of colorectal cancer screening below age 40, thereby removing screening as a source of bias," [lines 103-105]. Could this influence the results towards a bias for patients <40 years old to have MSI-H? In other words, it is possible that MSI-H tumors are more likely than MSS tumors to be discovered without screening, artificially boosting the frequency of <40 year olds having MSI-H tumors. Again, treating age as a numerical variable could assuage these concerns.

Response. We have addressed this point in several ways. In regard to identifying sources of bias for patients under 40 years old, we now note explicitly in the Discussion that because these patients are all stage IV, we would assume that screening would be a very limited factor in biasing the data, as the assumption would be that screening would allow earlier detection of malignancy. More importantly, we also have done as suggested and analyzed the data numerically, resulting in more informative analysis of age-mutation trends for a number of the analyses (see new panels in Figures 3, 4, and 5, and associated discussion). This has been extremely useful in reinforcing several of the conclusions made, and we thank the reviewer for the suggestion.

Point 3. The authors should state how they determined if a frequently mutated site on KRAS constituted as a "mutational hotspot." From Figure 2A, it seems that a site was deemed a hotspot if it appeared ≥ 10 times. In a population of 7,109 samples, this means the authors define a hotspot as a mutational frequency of ~0.14%, a threshold I am uncertain the rest of the field will agree with. Plotting the mutation burden instead of the frequency (as done in the rest of the paper) may be misleading. This is only magnified by displaying the y-axis on a logarithmic scale. If mutational frequency were plotted on a linear scale, codons mutated 10 times in a population of 7,109 would not likely be classified as high-frequency events.

Response. To determine if a frequently mutated site on the RAS proteins constitutes a “mutational hotspot”, we have used a binominal distribution model with a p-value cutoff 0.005. Thus, for KRAS the cutoff is 6 mutations/codon (p-value 0.00058); for NRAS, 4 mutations/codon (p-value 0.0012); and for HRAS, 4 mutations/codon (p-value 0.00068). While, indeed, using linear scale would allow the most frequent hotspots (i.e., G12 in KRAS) to completely dwarf all minor hotspots, we did not rely on visual assessment nor on an arbitrary cutoff to nominate hotspots. We have added text to the Methods section clarifying how hotspots were determined.

Point 4. It would be beneficial for the authors to more explicitly state which statistical tests they used for each figure. Further, the use of conditional inference trees or other more sophisticated and less common tools should be specifically justified over those simpler and more common. Additionally, the authors did not explain clearly why or when they used the Fisher exact test, z-ratio test, or the chi-squared test when they did. This was a confusing point as they are all tests for similar questions (are the proportions of a nominal variable different between two populations). Generally, it seems that the Fisher exact test is the most highly recommended. Finally, the figure legends do not explain the error bars in Figure 1C-E nor are any of the “ * ” symbols used to show statistical significance. It is not intuitive as to what they are describing and should be explicitly stated in the respective figure legends.

Response. We have modified the methods to indicate which statistical tests were used for the figures. Among these tests, conditional inference trees were used to simultaneously test many variants for association to patient characteristics. We chose this method to identify the strongest associations without over-fitting the model. For tests of frequencies/proportions, we now consistently use Fisher’s exact tests for all comparisons of proportions. In regard to the error bars in Figure 1C-E (and other figures), these show the 95% confidence intervals for the proportion of unstable tumors. We have added this explanation to the Figure legend, as requested.

Point 5. Do the authors adjust for multiple hypothesis testing throughout the manuscript? This is important to include in many of the figures to account for false positives. For example, in Figure 1C, there are four hypotheses tested: M–M, F–F, $F < 40 - M < 40$, and $F \geq 40 - M \geq 40$.

Response. We agree that multiple testing is an important consideration. As suggested by reviewer 1, and in response to this comment, we have therefore adopted the $\alpha = 0.005$ threshold recommended by Benjamin et al. Nat Hum Behav 2018 for declaring statistical significance. This stricter threshold will reduce the number of false-positive findings due to multiple testing. As a result, a few of the less important analyses that yielded significant differences based on the 0.05 criterion have lost significance; Figure panels reporting these data have been moved to Supplemental Figure S4. However, for the bulk of the work reported in this study, the conclusions hold.

Point 6. The authors should state how the lines in Figure 4E and F were constructed. Specifically, why are some non-linear where there does not appear to be a clear curvilinear relationship. This is important to prevent future readers from mis-interpreting the graphs.

Response We agree with the reviewer that there are not adequate data points to graph some relationships as curves. We have now consistently used linear models to indicate trends.

Minor Points

Point 1. A brief description of the technology used to gather the data at the heart of this study

included at the beginning of the results section would go a long way in framing the data for the reader.

Response. We have now begun the results section with the sentence, “Comprehensive genomic profiling of tumor samples by next generation sequencing (NGS) was performed in the course of routine clinical care for CRC patients with advanced disease, resulting in identification of genetic variants as well as biomarkers such as tumor mutation burden (TMB) and microsatellite stability.”

Point 2. Please include a comment on the nature and handling of samples from female or male discordant patients distinguished in Table 1 (e.g. are they included in figures comparing across sexes?).

Response. We have not included samples with discordant sex in any analyses comparing across sexes, and have added a line to the legend indicating this fact.

Point 3. In Figure 1A, is “FM” referring to “FMI” used elsewhere in the manuscript? Also, the legend “PAD, publicly available datasets” is covering part of the chart.

Response. We have standardized nomenclature to the term FM; we have also removed the statement about PAD from the Figure panel, as it is present in the legend. Thank you for catching these points.

Point 4. A higher-dimensional chart showing the information contained in Figure 1C-E may allow for easier comparison between all of the factors at hand.

Response. We experimented with a number of different ways to present the data. In the end, we believe the two-dimensional breakdown allows for easier comparison, particularly as we now provide two separate analyses comparing analyses for specimens originally designated MSI-H, and the expanded MSI-H set based on inclusion of samples above a threshold of total mutational burden. Hence, we have left the representation as we originally had it.

Point 5. There are a few instances of numbers without comma-separation every 3 digits to the left of the decimal point, while most do include the comma (eg. “1267 CRC tumors” [line 117] versus “~25,000 tumors of all types” [line 118]. Commas should be included as per Nature’s formatting standards.

Response. We have now carefully checked the manuscript and added comma separation.

Point 6. There are inconsistencies between the capitalization and italicization of KRAS, NRAS, and RAS. This should be standardized based on whether the use is in reference to the gene or protein.

Response. We have now carefully read through the manuscript and ensured consistent and appropriate nomenclature for gene and protein.

Point 7. On line 133, “Supplemental” is missing from “(Figure S2B, C).”

Response. Thank you, we have added this word.

Point 8. Figure 2A would benefit from a line showing the authors' threshold for "hotspot" classification (along with the points made in #3 of Major Points, above).

Response. We have added the requested line to Figure 2A (now Figures 2A-C, based on addition of new analyses).

Point 9. Figure 2F is missing the total number of tumor samples for MSI-H and MSS. This was a great addition in many of the other plots of similar format throughout the paper (including the adjacent panel G).

Response. We have now added the requested information to all panels in all Figures.

Point 10. By eye, there appear to be interesting correlations between Figure 3C and F and between Figure 3B and E. Seeing as the age of the patient and location of the tumor do not seem to be independent (Figure 1E), the graphing of location, age, and mutation of the codon (one chart for 12 and another for 61) could show some interesting associations.

Response. We agree that there are likely to be some interesting associations. Unfortunately, we were not able to find a good way to graph the interactions, given the complexity of the mutational profile. However, we have modified the discussion to bring out these issues more fully. In particular, we are intrigued by the pattern with mutations such as Q61K, which shows strong associations of age, gender, and location.

Point 11. Figure 3 B-F, H appear to be stretched or compressed, especially compared to panel A (this is an issue in other figures, too, though most noticeably here). Also, some of the numbers indicating sample size are out of place in the same panels.

Response. The original figures were slightly stretched as a way of adjusting visual alignment. We have now redrawn the figures to avoid the stretched effect, and also more correctly placed numbers indicating sample size.

Point 12. Is line 222 intended to say "KRAS" instead of "RAS"? Otherwise, the statement may need clarification.

Response. We had intended to say KRAS; this has been corrected.

Point 13. Some P values were not converted to scientific notation in cases where they should be (including on lines 225 and 244).

Response. Thank you for pointing this out. This has now been done.

Point 14. In my opinion, Figure 4E and F would be improved if reconfigured to adhere to more standard labeling practices. The label along the y-axes does not indicate what is measured. There are no labels along the x-axes. In addition, it does not appear that G12S, though included in the legend of panel F, is plotted in any of the graphs. Lastly, why are some of the lines dashed and others not in panel F? Is this intended to represent a property of the data? If so, it requires clarification in the legend.

Response. These are all excellent points. We have relabeled the X and Y axes in a number of the Figures to increase clarity. In regard to G12S, these data points were few in number, and

obscured by other lines on the plot. In regard to the dashed lines, those represented the less reliable trends with low counts. We have resolved both issues by combining all low-abundance G12 mutations into the category “G12 others” and using the same style as for other trends (uniformly non-dashed lines) for plotting (now in Figure 5G).

Point 15. There are two different spellings of “RASopathies” between lines 269 and 271.

Response. Apologies; this has been corrected.

Point 16. Please include a legend with the chart in Supplemental Figure 1. Also, if the authors continue to use the threshold of 16 mutations between MSS and MSI-H, it would be helpful to include a vertical line indicating this point on the chart.

Response. All legends are present in a submitted Supplemental Figure Legends File; we will work with the journal to ensure the legend appears adjacent to the Figure in the final article. We have also marked the point on the chart, as requested.

Reviewers' comments:

Reviewer #1 (Remarks to the Author):

The authors have successfully improved the manuscript addressing the points raised.

Reviewer #2 (Remarks to the Author):

Overall, the authors addressed most of my comments. A few are still outstanding, though should require very little effort to address.

Comments

1. In my previous Point 1, I inquire as to the final assignment of the 384 samples without TMB or MSS/MSI-H data. For unknown reasons, this are now only 267 samples in the current manuscript, but the final assignment is still not clear.

2. While the authors did append a brief description of the statistics behind assigning mutational hotspots, it is not obvious how they employed the model. For the sake of reproducibility, I request that the authors provide more detail than the single sentence they now include. My disagreement about the designation of "hotspot" to a mutation that occurs in 6 out of 13,336 patients ($6/13336 = 0.00045$) remains because this information is omitted. (To provide some perspective, according to gnomAD, there are 3 non-synonymous alleles at the KRAS locus that occur in the general population at the same frequency.) I am fine to disagree with the authors about the interpretation of the results, though I think it should be clear how the results were achieved.

3. The italicization of KRAS, NRAS, HRAS, and RAS are still inconsistent. While the gene names are general italicized, RAS is often left non-italicized. These inconsistencies extend to figures and figure legends.

4. The number of decimal places at which to use scientific notation is still inconsistent. For example, see lines 139 and 141 vs. 154 and 155.

Response to Reviewer #2:

Overall, the authors addressed most of my comments. A few are still outstanding, though should require very little effort to address.

1. In my previous Point 1, I inquire as to the final assignment of the 384 samples without TMB or MSS/MSI-H data. For unknown reasons, these are now only 267 samples in the current manuscript, but the final assignment is still not clear.

Response. We apologize for the lack of clarity. To address some of the large first round of critiques of this manuscript by the three reviewers, we worked extensively to mine additional data from the annotated records at Foundation Medicine. This resulted in the change in unassigned samples from 384 to 267. For almost all analyses in the study, these samples were excluded; the sole exception was for the global hotspot analysis reported in Figure 2. We have added text to the Legend to Table 1 to clarify this point.

2. While the authors did append a brief description of the statistics behind assigning mutational hotspots, it is not obvious how they employed the model. For the sake of reproducibility, I request that the authors provide more detail than the single sentence they now include. My disagreement about the designation of “hotspot” to a mutation that occurs in 6 out of 13,336 patients ($6/13336 = 0.00045$) remains because this information is omitted. (To provide some perspective, according to gnomAD, there are 3 non-synonymous alleles at the KRAS locus that occur in the general population at the same frequency.) I am fine to disagree with the authors about the interpretation of the results, though I think it should be clear how the results were achieved.

Response. We have now included a new Supplemental Data File 4 that includes counts of all mutations for each codon in each of the *RAS* genes in the analyzed data set, and calculation of probability for the number of each mutation at each codon; as well as comparison of the hotspots identified by this approach to the results reported by [Chang et al. 2017] (also published at www.cancerhotspots.org). Please see also the end of this document for a detailed description of the method employed; two independent biostatisticians agreed that this was a reasonable approach.

3. The italicization of KRAS, NRAS, HRAS, and RAS are still inconsistent. While the gene names are generally italicized, RAS is often left non-italicized. These inconsistencies extend to figures and figure legends.

Response. We have gone through the complete text, Figures, and Supplemental Data, and believe we have now consistently used italics for the gene, and non-italicized text for the protein, for each instance of KRAS, NRAS, HRAS, and RAS.

4. The number of decimal places at which to use scientific notation is still inconsistent. For example, see lines 139 and 141 vs. 154 and 155.

Response. We have reviewed all of the cited numbers, and believe we have effectively standardized the use of scientific notation in the study.

Extended Response to Point 2:

We have used an ad hoc approach analyzing the distribution of mutations on the linear RAS protein structure, as described below using KRAS as an example.

Essentially, for a ranked list of all KRAS mutations,

1) Given a total of 7049 KRAS mutations in our set, and 189 aa (length of KRAS) we calculated the probability of having 4757 (or more) mutations in any one codon by chance alone. This probability is well below 0.005, so the most frequent mutation (G12) is nominated as a hotspot.

2) With $7049 - 4757 = 2292$ mutations, and $189 - 1 = 188$ positions remaining, we again calculated the probability of having 1196 (or more) mutations in any one codon by chance alone.

3) We iteratively repeat this procedure until the probability of having X mutations in one of the Y remaining aminoacids exceeds our cutoff value 0.005. Thus, all positions identified in the previous steps are nominated as hotspots.

The calculation of probability was done using the cumulative binominal distribution

$E(x; n, p) = \sum_{y=0}^x b(y; n, p)$ where x = number_of mutations in a given codon, n = number of all mutations not yet assigned to the previously defined hotspots, and p (probability of mutation) is calculated as $1/(\text{number of the codons under consideration})$.

Calculations were implemented in the Excel spreadsheet, which we provide as a supplementary file.

REVIEWERS' COMMENTS:

Reviewer #2 (Remarks to the Author):

All of my concerns have been addressed.